# Increased Surgical Experience in Microendoscopic Spinal Surgery Can Reduce Development of Postoperative Spinal Epidural Hematoma and Improve the Clinical Outcomes

**DOI:** 10.3390/jcm11216495

**Published:** 2022-11-01

**Authors:** Masayoshi Iwamae, Koji Tamai, Kunikazu Kaneda, Hidetomi Terai, Hiroshi Katsuda, Nagakazu Shimada, Hiroaki Nakamura

**Affiliations:** 1Department of Orthopaedic Surgery, Osaka Metropolitan University Graduate School of Medicine, Osaka 545-8585, Japan; 2Department of Orthopaedic Surgery, Shimada Hospital, Osaka 583-0875, Japan

**Keywords:** postoperative spinal epidural hematoma, microendscopic surgery, acute phase MRI, risk factor, surgical technique, clinical outcome, quality of life

## Abstract

No reports have previously evaluated the association between surgical technique and the incidence of postoperative spinal epidural hematoma (PSEH) following microendoscopic decompression surgery (MED). This study aimed to evaluate the association between the development of radiographic PSEH (rPSEH) following MED and microendoscopic surgical experience and postoperative clinical outcomes related to the quality of life (QoL). This retrospective cohort study included 3922 patients who had undergone MED performed by a single surgeon. rPSEH was defined as a hematoma that was identified via routine magnetic resonance images performed 3–4 days postoperatively. Patients were divided into rPSEH and control groups to identify the risk factor of rPSEH and assess clinical outcomes. In the multivariate analysis, age (*p* = 0.002), surgical experience (*p* = 0.003), surgical time (*p* = 0.038), multilevel decompression (*p* < 0.001), and diagnosis (*p* = 0.004) were identified as independent variables associated with rPSEH. Moreover, in mixed-effect models, the rPSEH group showed less improvement in Oswestry Disability Index (*p* = 0.014) than the control group. In conclusion, the surgical experience was identified as a risk factor for rPSEH that could lead to poor QoL. The sharing of microendoscopic surgical techniques among surgeons may reduce rPSEH incidence and improve patients’ QoL.

## 1. Introduction

Postoperative spinal epidural hematoma (PSEH) is one of the most common complications of spinal surgery. It sometimes causes intractable pain and progressive neurological deficit, including motor weakness and/or bowel bladder dysfunction, which require the emergent surgical evacuation of the hematoma [1,2]. Several reports have shown that PSEH incidence detected using magnetic resonance imaging (MRI) ranges from 15% to 89% [3,4]. Interestingly, a previous report demonstrated that even asymptomatic PSEH could lead to poor clinical improvement [3]. To date, several previous reports identified risk factors of PSEH, including high body mass index, a history of hypertension, lumbar hypolordosis, and estimated blood volume eliminated via the drainage tube [2,5,6,7]. The majority of the risk factors identified were derived from patient characteristics and were not under the control of the spinal surgeon. In order to improve surgical outcomes, it is important to identify risk factors for PSEH that spinal surgeons are able to minimize.

Microendoscopic surgery in spinal surgery for lumbar disc herniation (LDH) was first reported by Foley and Smith in 1997, and its application in lumbar spinal stenosis (LSS) was reported by Ikuta et al. in 2001 [8,9]. Microendoscopic surgical techniques were then established as a standard choice for treating LDH and LSS, with good outcomes [10,11,12]. Meanwhile, some reports suggested the presence of a learning curve for performing microendoscopic decompression surgery [9,10,13,14].

We hypothesized that there might be a relationship between the surgical experience of a surgeon who performs microendoscopic surgery and the incidence of PSEH. However, to our knowledge, no reports have previously evaluated the association between surgical technique and the incidence of PSEH. This is likely because such a study would be difficult to design. At our institution, all patients who underwent microendscopic decompression were treated by a single surgeon using the same standardized care protocols, in which all patients received a lumbar MRI 3–4 days after the surgery in order to check whether the nerve root or dura matter was decompressed after surgery regardless of the existence of symptoms. Therefore, we were able to use our database to investigate the association between surgical experience and the incidence of radiographic PSEH (rPSEH). Hence, the primary aim of this study was to identify the factors associated with rPSEH after microendoscopic decompression surgery, including the surgical experience of a single surgeon. In addition, we evaluated the effect of rPSEH on surgical outcomes related to the quality of life (QoL) to determine the importance of preventing rPSEH.

## 2. Materials and Methods

This was a retrospective cohort study. All study participants provided informed consent, and an MRI consent form was obtained after surgery. The study protocol was approved by the Institutional Review Board of our institution. No funds were received in support of this work.

### 2.1. Patient Population

We prospectively collected data from 4199 consecutive patients who underwent microendoscopic decompression surgery for LDH and LSS performed by a single surgeon at our institution between 2011 and 2019 and were followed up for at least 3 months postoperatively. We excluded the 277 patients that underwent repeat procedures. In total, 3922 patients were enrolled in the analysis (1415 females, 2507 males; mean age at surgery, 55.4 ± 18.0 years; 2032 LDH patients, 1890 LSS patients).

### 2.2. Surgical Criteria

Microendoscopic posterior decompression was indicated in patients with neurogenic claudication or radicular pain associated with neurological signs and stenosis or herniation consistent with symptoms observed via MRI, which did not improve despite the application of conservative treatment for at least three months. Exclusion criteria included patients with degenerative spondylolisthesis of more than grade 2, spondylolytic–spondylolisthesis, and degenerative lumbar scoliosis with a Cobb angle of > 20°.

### 2.3. Perioperative Clinical Care

All patients who underwent microendoscopic decompression were treated according to the same standardized care protocols of our institution. Decompression surgery was performed using the same procedure previously reported [15]. Patients were placed under general anesthesia, and a postoperative closed-suction drain (silicone, round, 10 French gauge, drainage tube type; Tokibo Co., Ltd.; Tokyo, Japan) was placed in the extradural space before closure. All patients were allowed to sit and walk with a soft brace on the day after the surgery. The suction drain was removed 24 h after surgery in LDH cases and 48 h after surgery in LSS cases. The care protocol included the routine use of 200 mg/day cerecoxib up to seven days after the surgery and allowed patients to use additional painkillers such as acetaminophen, opioids, or non-steroidal anti-inflammatory drugs with oral or intravenous administration methods as needed. In addition, all patients received a lumbar MRI three to four days after surgery, which was defined as an acute-phase MRI in this study. It was recommended that patients be discharged from the hospital six or seven days postoperatively.

### 2.4. Definition of rPSEH

The presence of an rPSEH was evaluated using axial T2 weighted images on either a 1.5-T MRI (Signa^TM^ Creator; GE Healthcare, Tokyo, Japan) or a 3.0-T MRI (Signa^TM^ Pioneer; GE Healthcare, Tokyo, Japan). In this study, rPSEH was defined as a hematoma that compressed the dura mater on axial MRI regardless of the presence or absence of symptoms (Figure 1). Two observers (MI and KT) independently evaluated all MRIs, and differences between determinations made by the observers were settled by reaching a consensus. The kappa coefficient within observers after a one-month interval was 0.840 (*p* < 0.001), and between the observers, it was 0.727 (*p* < 0.001).

### 2.5. Preoperative Data

From the medical record of patients, information regarding patient age at the time of surgery, sex, height, weight, body mass index (BMI), and the number of preoperative epidural injections were collected. In terms of preoperative symptoms, the presence of motor weakness and bowel bladder syndrome was noted.

### 2.6. Surgical Data

The accumulated number of surgeries performed by the surgeon determined the surgical experience. Surgical data, including the total duration of surgery and the number of surgical segments, were collected. The duration of surgery per segment was calculated by dividing the total duration of surgery by the number of surgical segments. In addition, the number of cases with a dural puncture during surgery was recorded. Postoperative surgical data collected included the duration of the hospital stay and the use of additional painkillers. When the patient used a pain killer other than celecoxib, which was routinely prescribed, the number of times it was used was recorded.

### 2.7. Clinical Score

Japanese Orthopedics Association (JOA) scores for degenerative lumbar disease and Oswestry Disability Index (ODI) values were collected both preoperatively and three months postoperatively [16,17].

### 2.8. Statistical Analyses

Firstly, the incidence rate of rPSEH was investigated every 1,000 cases. The previous reports particularly showed that 20–30 cases were required complete the learning curve of microendoscopic surgery for LDH or LSS [13,14]. Therefore, in addition, the incidence rates of rPSEH in the learning period (1–25 cases), proficient period (26–100 cases), and expert period (101–1000 cases) were investigated, respectively.

Secondly, all patients were divided into two groups according to MRI findings: an rPSEH group and a control group. In order to identify factors associated with rPSEH, all preoperative background data, surgical data, and clinical scores for the two groups were compared using a *t*-test or chi-squared test, as appropriate. Comparisons of variables producing *p*-values less than 0.05 via previously performed univariate analyses were included as explanatory variables in the multivariate logistic regression analysis. The objective variable was the presence of rPSEH. Adjusted odds ratios (aORs), *p*-values, and 95% confidence intervals (95% CI) were calculated.

Thirdly, the influence of rPSEH on the postoperative course, postoperative additional pain killer use, and duration of hospital stay were compared using *t*-tests. Improvements in clinical scores observed for each group were compared using a mixed-effects model that was adjusted for age, diagnosis, and the number of surgical segments.

All analyses except for the mixed-effects model were performed using the R program (version 3.5.2; R Foundation for Statistical Computing, Vienna, Austria) and EZR computer software (Saitama Medical Center, Jichi Medical University, Saitama, Japan) [18]. The mixed-effects model was calculated using SPSS software (SPSS version 23; IBM, Chicago, IL, USA). Values of *p <* 0.05 were considered statistically significant.

## 3. Results

### 3.1. Incidence Rate of rPSEH

Among the 3922 patients assessed, rPSEH was identified in 203 individuals (5.2%) via acute phase MRI. These patients were placed within the rPSEH group (Figure 1). Of these, four (0.1%) required the emergent surgical evacuation of the hematoma. The incidence of rPSEH was highest in the first 1,000 cases (7.0%) assessed, followed by the second and third sets of 1,000 cases (4.9% and 4.8%, respectively). The lowest incidence of rPSEH was observed within the most current 922 cases (3.9%). Moreover, the incidence rates of rPSEH in the learning period (1–25 cases), proficient period (26–100 cases), and expert period (101–1000 cases) were 28.0%, 4.0%, and 6.7%, respectively. Therefore, the results showed that the incidence rate of rPSEH was significantly higher than that in the other periods (*p* < 0.001, Table 1).

### 3.2. Risk Factors of rPSEH

Univariate analysis of preoperative data revealed that the average age (*p* < 0.001), BMI (*p* < 0.001), and pre-op JOA score (*p* = 0.044) were higher, and the occurrence of preoperative motor weakness was lower (*p* = 0.018) in the rPSEH group than in the control group (Table 2). Regarding surgical factors, less surgical experience (*p* = 0.017), increased LSS incidence (*p* < 0.001), increased number of surgical segments (*p* < 0.001), and longer duration per segment (*p* < 0.001) were observed in the rPSEH group than in the control group (Table 1 and Table 3). A subsequent multivariate logistic regression analysis showed that age (aOR = 1.87, *p* = 0.002); accumulated numbers of surgeries, in other words, surgical experience (aOR = 1.62, *p* = 0.003); occurrence of LSS versus LDH (aOR = 1.93, *p* = 0.004); surgical time per segment (aOR = 1.44, *p* = 0.038); and the occurrence of multilevel surgery (aOR = 1.74, *p* < 0.001) were determined to be dependent variables that significantly determined the risk of rPSEH identified via acute phase MRI (Table 4).

### 3.3. Clinical Outcomes

The period hospitalized was longer (7.4 vs. 6.4 days; *p* < 0.001), and additional painkiller use was higher (1.20 ± 1.16 times vs. 1.05 ± 0.91 times; *p* = 0.023) in the rPSEH group than in the control group (Table 3). JOA score and ODI values were significantly improved at final-follow up versus preoperatively for both groups (*p* < 0.001). However, for the mixed-effects model, which was adjusted for age, diagnosis, and the number of surgical segments required, the rPSEH group showed significantly less improvement in the JOA score and ODI value than did the control group (*p* = 0.037 and *p* = 0.014, respectively; Table 5).

## 4. Discussion

In our datasets, rPSEH incidence on acute phase MRI was 5.2%. The current study revealed that independent risk factors of rPSEH were patient age, LSS versus LDH surgery, surgical duration per segment, multilevel decompression, and the surgeon’s microendscopic surgery experience. In addition, rPSEH significantly affected the in-hospital period, additional painkiller use, and postoperative clinical score. JOA score and ODI were negatively associated with rPSEH incidence.

The rate of PSEH detected by MRI ranged from 15% to 89%, and PSEH incidence in our study was lower than the rates previously reported [3]. This may have occurred because all patients included in our study received a routine MRI postoperatively. In previous studies, almost all patients receiving MRI after surgery had previously experienced symptoms, including low back pain or progressive neurological deficits. As a result, these reports had a high incidence of PSEH in comparison with the current study. To our knowledge, only one study examined patients (30 consecutive) who had received a routine lumbar MRI regardless of symptoms [3]. In that report, Ikuta et al. reported that the incidence rate of rPSEH was 33% via lumbar MRI a week after microendoscopic surgery. Since the number of subjects in the current study (*n* = 3922) was much larger than that of the previous report, the rPSEH incidence rate reported here (5.2%) after microendoscopic decompression surgery is likely more accurate than that reported by Ikuta et al.

Fujita et al. reported that JOA and VAS values were significantly worse in patients with PSEH than controls at discharge [2]. Moreover, Ikuta et al. showed that patients with PSEH had higher postoperative VAS values for lower back pain, used analgesics more frequently, and had longer hospital stays than patients without PSEH [3]. The results of this study are in accordance with these published results. Although our definition of rPSEH did not consider the existence of clinical symptoms, rPSEH incidence had a negative effect on multiple types of clinical outcomes. One reason for this phenomenon may be the release of several cytokines from the hematoma. Previous evidence suggested that cytokines released from a hematoma after trauma or fracture could accelerate inflammation and fibromatosis, resulting in joint contracture or muscle stiffness [19]. rPSEH after microendoscopic decompression surgery should also increase cytokine concentrations, which has the potential to cause inflammation and fibromatosis and result in low back pain immediately after surgery and/or postoperative lumbar stiffness [20]. Furthermore, this mechanism may explain the findings of the current study in which patients with rPSEH showed less improvement in the JOA score and ODI at a final follow-up. A previous report particularly demonstrated that ODI and both physical component summary (PCS) and mental component summary (MCS) of short form 36 (SF-36) health survey had a significant correlation [21]. Therefore, preventing the development of rPSEH could improve the patients’ health-related quality of life (Hr-QoL).

Some of the risk factors for rPSEH identified in our study, such as high age, LSS versus LDH, surgical duration per segment, and the occurrence of multilevel surgery, have already been demonstrated in previous reports [2,3,22,23]. The key finding of the current study, which was previously reported, suggested that surgical experience was the independent factor associated with rPSEH incidence. It is well-known that, as they accumulate surgical experience, surgeons correspondingly gain knowledge useful for performing surgeries more effectively. Key strategies the surgeon who performed procedures in this study determined to be important for reducing rPSEH incidence were stopping minor bleeding by pressing bone wax into the bleeding cancellous bone or by coagulation of the vessel on and around the dura using bipolar coagulator without dural heat-injury [24]. Use of these intraoperative tips for stopping bleeding may require surgical skill and experience; however, we believe that the sharing of these key strategies has the potential to help all spinal surgeons, especially a surgeon in the learning period, achieve improved postoperative outcomes.

In this study, we evaluated rPSEH, which is a preoperative hematoma found on MRI regardless of symptoms. Current findings do not suggest a need for physicians to take acute-phase MRIs for all cases. The benefits of taking routine MRIs may be outweighed by concerns regarding medical costs since just 0.1% of patients included within our dataset required emergency surgery. However, current results suggest that the prevention of rPSEH is critical because it significantly worsens clinical outcomes after surgery. In addition, this work suggests that sharing surgical knowledge, particularly regarding strategies needed to stop bleeding during surgery, may reduce the occurrence of rPSEH since we verified that surgical experience is an independent factor associated with rPSEH incidence.

There are several limitations to the present study. First, postoperative care, including the in-hospital period, is not permitted by insurance systems used in countries such as the United States; however, the Japanese system allowed us to assess these parameters. Despite its limitations, we believe that current results may be beneficial to physicians globally. Second, the clinical scores were assessed three months postoperatively. Third, there are individual differences in the learning curves of microendoscopic surgery. A validation of this study that includes an assessment of patients who undergo microendscopic surgery performed by another surgeon is needed to ensure that the findings reported here may be generalized. Fourth, it might be difficult to categorize the postoperative hematoma as a natural history or a complication. That is why we defined the hematoma as “radiographic” PSEH in this study. However, we think it is worth reducing the development of radiographic PSEH because our results demonstrated that even radiographic PSEH negatively affected the improvement of clinical outcomes. Finally, the retrospective nature of the study makes it difficult to completely exclude the possibility of selection and surgeon bias. In order to overcome the limitations mentioned, a large-scale prospective study with a long period of rPSEH patient follow-up should be carried out. A strength of this study is that all cases were treated by a single surgeon at one institution, which may reduce surgical indication, method, and skill bias and minimize differences regarding pre- and postoperative care. Further, detailed preoperative data were provided. To our knowledge, this is the first report involving a large number of cases in which all patients who underwent microendoscopic surgery were evaluated using acute phase MRI. Therefore, in spite of its limitations, we believe current study findings may provide surgeons with knowledge beneficial for preventing rPSEH and, therefore, improving clinical outcomes.

## 5. Conclusions

In conclusion, the incidence rate of rPSEH on acute phase MRI was 5.2%, and rPSEH incidence negatively affected the in-hospital period, additional painkiller use, and clinical score improvement. This study revealed that one of the factors associated with rPSEH incidence on acute phase MRI was the surgeon’s microendoscopic surgical experience. Based on these findings, we believe that sharing surgical knowledge may reduce rPSEH incidence, which has the potential to improve patients’ Hr-QoL.

## Figures and Tables

**Figure 1 jcm-11-06495-f001:**
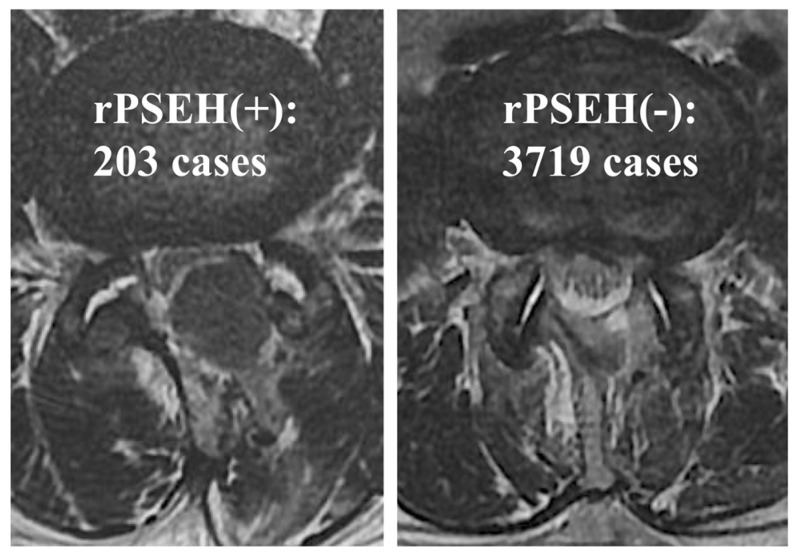
Postoperative magnetic resonance image of representative case with and without postoperative spinal epidural hematoma. Although the left case with rPSEH had had severe buttock pain postoperatively, the pain gradually disappeared without emergent surgical evacuation of the hematoma. rPSEH; radiographic postoperative spinal epidural hematoma.

**Table 1 jcm-11-06495-t001:** The incidence rate of radiographic postoperative spinal epidural hematoma based on the number of surgical experiences.

			rPSEH Group (*n* = 203)	Control Group (*n* = 3719)	*p*-Value
Accumulated surgical experiences (cases)		1–1000	70 (7.0%)	930	0.017
		1001–2000	49 (4.9%)	951	
		2001–3000	48 (4.8%)	952	
		3001–3922	36 (3.9%)	886	
Initial 1000 cases	Learning period	1–25	7 (28.0%)	18	<0.001
	Proficient period	26–100	3 (4.0%)	72	
	Expert period	101–1000	60 (6.7%)	840	

**Table 2 jcm-11-06495-t002:** Comparison of patients’ demographic data between the patients with and without rPSEH.

		rPSEH Group (*n* = 203)	Control Group (*n* = 3719)	*p*-Value
Age (years old)		64.9 ± 14.7	54.8 ± 18.0	<0.001
Sex	Male	138	2369	0.245
	Female	65	1350	
BMI (kg/m^2^)		25.1 ± 3.7	23.9 ± 3.7	<0.001
Pre-op JOA score		14.3 ± 4.9	13.5 ± 5.3	0.044
Pre-op ODI		41.8 ± 16.3	44.2 ± 19.5	0.118
The number of pre-op epidural injection		1.2 ± 1.8	1.0 ± 1.9	0.343
Pre-op motor weakness	MMT = 5	187	3228	0.018
	MMT ≤ 4	14	479	
Pre-op bowel bladder disfunction	(−)	201	3685	0.541
	(+)	0	22	

rPSEH—radiographic postoperative spinal epidural hematoma; BMI—body mass index; pre-op—preoperative; JOA—Japanese Orthopaedic Association; ODI—Oswestry Disability Index; statistical analysis: Student’s *t*-test or chi-square test.

**Table 3 jcm-11-06495-t003:** Comparison of surgical data between the patients with and without rPSEH.

		rPSEH Group (*n* = 203)	Control Group (*n* = 3719)	*p*-Value
Diagnosis	LSS	159	1731	<0.001
	LDH	44	1988	
The number of surgical segments	1 level	129	3122	<0.001
	2 levels	61	518	
	3 levels	13	78	
	4 levels	0	1	
Surgical time per segment (min)		82.3 ± 27.0	69.4 ± 26.1	<0.001
Intraoperative dural puncture (cases)	(−)	195	3639	0.251
	(+)	8	75	
In-hospital period (days)		7.4 ± 3.4	6.4 ± 2.2	<0.001
Additional painkiller use (times)		1.2 ± 1.2	1.1 ± 0.9	0.023

rPSEH—radiographic postoperative spinal epidural hematoma; LSS—lumbar spinal stenosis; LDH—lumbar disc herniation; statistical analysis: Student’s *t*-test or chi-square test.

**Table 4 jcm-11-06495-t004:** Multivariate linear regression analysis (objective variables: postoperative spinal epidural hematoma).

Explanation Variables		Reference	aOR	*p*-Value	95%CI
Age	(55 years)	≤54	1.87	0.002	1.25–2.81
BMI	(≥25 kg/m^2^)	<25	1.33	0.056	0.99–1.79
Surgical experiences	(1–1000 cases)	≥1001	1.62	0.003	1.19–2.22
Diagnosis	(LSS)	LDH	1.93	0.004	1.24–3.01
Surgical time per segment	(≥70 min)	<70	1.44	0.038	1.02–2.04
Surgical segments	(multilevel)	One level	1.74	<0.001	1.25–2.41
Pre-op JOA score	(≥14 points)	<14	1.07	0.649	0.80–1.44
Pre-op motor weakness	(−)	(+)	1.35	0.301	0.76–2.39

rPSEH—radiographic postoperative spinal epidural hematoma; aOR—adjusted odds ratio; CI—confident interval; BMI—body mass index; pre-op—preoperative; LSS—lumbar spinal stenosis; LDH—lumbar disc herniation; JOA—Japanese Orthopaedic Association.

**Table 5 jcm-11-06495-t005:** Clinical scores compared between the patients with and without rPSEH using mixed-effects model.

Clinical Scores	rPSEH Group (*n* = 203)	Control Group (*n* = 3719)	*p*-Value
JOA score			0.037
Preoperative	14.3 ± 4.9	13.5 ± 5.3	
Postoperative	26.4 ± 2.4	26.6 ± 2.5	
ODI			0.014
Preoperative	41.8 ± 16.3	44.2 ± 19.5	
Postoperative	9.5 ± 12.0	7.2 ± 9.7	

rPSEH—radiographic postoperative spinal epidural hematoma; JOA—Japanese Orthopaedic Association; ODI—Oswestry Disability Index; statistical analysis: mixed-effects model.

## Data Availability

The datasets generated and/or analyzed during the current study are available from the corresponding author on reasonable request.

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
