# Peer review of "Increased Surgical Experience in Microendoscopic Spinal Surgery Can Reduce Development of Postoperative Spinal Epidural Hematoma and Improve the Clinical Outcomes"

_jcm, 2022, doi:10.3390/jcm11216495_

Round 1
Reviewer 1 Report
Although of retrospective design, the authors have extensively analyzed the factors associated with the development of postoperative spinal epidural hematoma in a multivariate model. It is a well written manuscript. I would suggest the authors cut-short the 'introduction' section, and make it more brief and relevant to the current context of the study.
Author Response
Reviewer 1
Although of retrospective design, the authors have extensively analyzed the factors associated with the development of postoperative spinal epidural hematoma in a multivariate model. It is a well written manuscript. I would suggest the authors cut-short the 'introduction' section, and make it more brief and relevant to the current context of the study.
Response:
Thank you very much for your careful review and thoughtful comment. According to your comments, we have revised our manuscript to make it more briefly and logically to the context of the study.

Reviewer 2 Report
This paper is based radiographic epidural haematoma. Can the authors identify if this is part of the natural history after the intervention or are they classifying this as a complication.What was the purpose of performing a routine MRI after an endoscopic operation. Was this an ethics approved intervention, or I this your clinical practice. The start of the introduction states that the hepatoma sometimes causes pain and neurological deficit. Clearly, there will be some that are asymptomatic. What was the purpose of an intervention such as MRI scan on such patients. Did you perform an MRI on all 4.199 patients 4 days after the operation. We’re they consented for this? Was the patient in Figure 1 symptomatic or asymptomatic? Please clarify these points for a decision on submission.
Author Response
Reviewer 2
This paper is based radiographic epidural haematoma. Can the authors identify if this is part of the natural history after the intervention or are they classifying this as a complication.
Response:
Thank you very much for your suggestion. It might be difficult to categorize the postoperative hematoma as a natural history or a complication. However, we think it is important to reduce the development of radiographic postoperative spinal epidural hematoma because our results demonstrated that the hematoma negatively affected the improvement of ODI or JOA score. According to this comment, we realized that this would be one of our important limitations. Hence, we added the description regarding to this issue in the limitation section. (line 306-310)
What was the purpose of performing a routine MRI after an endoscopic operation.
Response:
In our clinical settings, we aimed to evaluate the decompression of nerve roots and dura mater after surgery by routine MRI.
Was this an ethics approved intervention, or I this your clinical practice.
Response:
Thank you very much for this question. We analyzed the prospectively collected data retrospectively. We have obtained approval from our institutional review board for this study and all patients provided the informed consent. (Line 66-68)
The start of the introduction states that the hepatoma sometimes causes pain and neurological deficit. Clearly, there will be some that are asymptomatic. What was the purpose of an intervention such as MRI scan on such patients. Did you perform an MRI on all 4.199 patients 4 days after the operation. We’re they consented for this?
Response:
As you pointed out, some of our patients were asymptomatic. We agree with you that “routine MRI after surgery” might be the issue of the burden on the medical economy for insurance company and patients. However, under the Japanese insurance system, MRI during hospitalization was completely covered by the insurance. Hence, taking MRI during hospitalization do not require the extra fee from patients. Therefore, the MRI consent form could be obtained from all the patients, and we added this sentence (line 67). This is the reason that we have been able to perform MRI imaging routinely. Taking different perspective, we believe that the Japanese insurance system made this study possible. We would like to stress that current results do not encourage all surgeons to take MRI after surgery routinely. However, our results could provide us with new findings regarding to the hematoma after surgery.
Was the patient in Figure 1 symptomatic or asymptomatic? Please clarify these points for a decision on submission.
Response:
The postoperative spinal epidural hematoma showed in Figure 1 was symptomatic with buttock pain. Fortunately, however, the symptoms were gradually disappeared without emergent surgery to remove the hematoma (line 179-181).

Reviewer 3 Report
Interesting paper on how surgical experience of microendoscopic spinal surgery can reduce risks of spinal epidural haematoma. It is also interesting that the authors managed to get a regular MRI scan 3-4 days post-operatively for all patients including those without symptoms.
All surgeries were performed by a single surgeon and it is an excellent way to show the learning curve. However, different people may learn at a different rate and may not be representative of all surgeons. Patient that also underwent repeat procedures would actually count as multiple folds of experience points for the surgeon BUT they were not included in the study. If these cases contributed to the surgeon's experience, they can be included in the first 1000 cases (or 1001-2000 cases) but excluded in terms of calculation for post-op spinal epidural haematoma. The method of deciding how the cut offs (of 1-1000, 1001-2000 etc) of the groups were devised would also be interesting. Perhaps they can be group as there have been previous studies suggesting that the learning curve of this procedure is about 20+, and therefore, comparing between learning (0-25 cases) vs proficient (25-100 cases) vs expert (>100 cases) would also be beneficial.
Definitely, surgical experience does play a role - in how well surgeons are able to avoid venturing to areas of bleeding/using bone wax or bipolar coagulator to stop bleeding. But, as the study looks at the rate of post-op spinal epidural haematoma, other factors that may contribute should also be included. The biggest factor would be the patient's comorbidities, use of anticoagulants, pre-op coagulation factors, operative duration, use of coagulative agents (such as Floseal) etc.
Overall, it is a good paper to show that there continues to be measurable improvements even after 1000 cases and experience plays a big role in reducing peri-operative complications.
Author Response
Reviewer 3
Interesting paper on how surgical experience of microendoscopic spinal surgery can reduce risks of spinal epidural haematoma. It is also interesting that the authors managed to get a regular MRI scan 3-4 days post-operatively for all patients including those without symptoms.
Response: Thanks for your kindly words and appropriate summary.
All surgeries were performed by a single surgeon and it is an excellent way to show the learning curve. However, different people may learn at a different rate and may not be representative of all surgeons. Patient that also underwent repeat procedures would actually count as multiple folds of experience points for the surgeon BUT they were not included in the study. If these cases contributed to the surgeon's experience, they can be included in the first 1000 cases (or 1001-2000 cases) but excluded in terms of calculation for post-op spinal epidural haematoma. The method of deciding how the cut offs (of 1-1000, 1001-2000 etc) of the groups were devised would also be interesting. Perhaps they can be group as there have been previous studies suggesting that the learning curve of this procedure is about 20+, and therefore, comparing between learning (0-25 cases) vs proficient (25-100 cases) vs expert (>100 cases) would also be beneficial.
Response:
Thanks for your very important comments. We completely agree with your that there are individual differences in learning curves that could not be demonstrated in this study. Moreover, the experience of reoperation cases was excluded in this study. Hence, we have emphasized the sentence in the limitation section. (line 302-303) According to your suggestion, additional analyses were performed separately for learning (1-25 cases) vs proficient (26-100 cases) vs expert (>100 cases). the results have showed that the incidence of radiographic PSEH was 7/25 (28%) for learning period (1 to 25 cases), 3/75 (4%) for proficient period (26 to 100 cases), and 60/1000 (6.0%) at expert period (101 to 1000 cases). That indicated that the incidence of radiographic PSEH was significantly higher in the learning period (chi-square test; p<0.001). We have added the interesting results to the main text (line 142-146, 172-176, 287) and Table 2, which can emphasize the significance of this study. Again, we appreciate this reviewer for giving us this a chance to improve our manuscript.
Definitely, surgical experience does play a role - in how well surgeons are able to avoid venturing to areas of bleeding/using bone wax or bipolar coagulator to stop bleeding. But, as the study looks at the rate of post-op spinal epidural hematoma, other factors that may contribute should also be included. The biggest factor would be the patient's comorbidities, use of anticoagulants, pre-op coagulation factors, operative duration, use of coagulative agents (such as Floseal) etc.
Response:
In this retrospective study, we do not have the information about the use of anticoagulants and preoperative coagulation factors. Hence, we enforced the weakness of retrospective study in the limitation section. (line 310-312) As coagulative agents such as Floseal was not used in any case during the study period. Therefore, we do not demonstrate the effect of coagulative agents on the prevention of radiographic PSEH.
Overall, it is a good paper to show that there continues to be measurable improvements even after 1000 cases and experience plays a big role in reducing peri-operative complications.
Response:
We gratefully appreciate you for your understanding of our study. Your comments have made this paper much more clinically significant.
